# Lack of ownership of mobile phones could hinder the rollout of mHealth interventions in Africa

Justin T Okano[1], Joan Ponce[1], Matthias Krönke[2], Sally Blower[1]*

[1]Center for Biomedical Modeling, Department of Psychiatry and Biobehavioral Sciences, Semel Institute for Neuroscience and Human Behavior, David Geffen School of Medicine, University of California, Los Angeles, Los Angeles, United States; [2]Afrobarometer / Institute for Democracy, Citizenship and Public Policy in Africa, University of Cape Town, Cape Town, South Africa

**Abstract** Mobile health (mHealth) interventions, which require ownership of mobile phones, are being investigated throughout Africa. We estimate the percentage of individuals who own mobile phones in 33 African countries, identify a relationship between ownership and proximity to a health clinic (HC), and quantify inequities in ownership. We investigate basic mobile phones (BPs) and smartphones (SPs): SPs can connect to the internet, BPs cannot. We use nationally representative data collected in 2017–2018 from 44,224 individuals in Round 7 of the Afrobarometer surveys. We use Bayesian multilevel logistic regression models for our analyses. We find 82% of individuals in 33 countries own mobile phones: 42% BPs and 40% SPs. Individuals who live close to an HC have higher odds of ownership than those who do not (aOR: 1.31, Bayesian 95% highest posterior density [HPD] region: 1.24–1.39). Men, compared with women, have over twice the odds of ownership (aOR: 2.37, 95% HPD region: 1.96–2.84). Urban residents, compared with rural residents, have almost three times the odds (aOR: 2.66, 95% HPD region: 2.22–3.18) and, amongst mobile phone owners, nearly three times the odds of owning an SP (aOR: 2.67, 95% HPD region: 2.33–3.10). Ownership increases with age, peaks in 26–40 year olds, then decreases. Individuals under 30 are more likely to own an SP than a BP, older individuals more likely to own a BP than an SP. Probability of ownership decreases with the Lived Poverty Index; however, some of the poorest individuals own SPs. If the digital devices needed for mHealth interventions are not equally available within the population (which we have found is the current situation), rolling out mHealth interventions in Africa is likely to propagate already existing inequities in access to healthcare.

*For correspondence: sblower@mednet.ucla.edu

Competing interest: The authors declare that no competing interests exist.

## Editor's evaluation

This study used 2017-2018 Afrobarometer surveys of nearly 45,000 individuals to examine the association between the ownership of mobile phones and proximity to a health clinic in 33 African countries. Findings show that about 40% of people own smartphones and those who live closer to health clinics are more likely to own a mobile phone. This manuscript will be of interest to all people who are involved in the design and implementation of mHealth interventions in Africa.

## Introduction

The effectiveness of mobile health (mHealth)-based interventions is currently under investigation in many countries in Africa (*Holst et al., 2020*; *Manby et al., 2022*; *Onukwugha et al., 2022*; *Osei et al., 2020*; *Wood et al., 2019*). The objective of these interventions is to increase access to healthcare

**eLife digest** Many healthcare systems in African countries are under-resourced. As a result, people, particularly those living in rural areas, often have to travel large distances to access the medical care they need. Mobile phone-based interventions (also known as mHealth) could make a substantial difference. In Africa, mHealth is already used to diagnose and treat diseases, increase adolescents' use of sexual and reproductive health services, boost HIV prevention and treatment, and improve maternal and child healthcare.

However, using mHealth services requires owning a basic mobile phone or, in some cases, a smartphone that can access the internet. While mobile phone ownership in Africa is increasing rapidly, data on who has them and what types of phones they have are limited. If geographic, income, or gender-based inequities exist, mHealth interventions may not be able to reach those who would benefit the most.

To close this knowledge gap, Okano et al. analyzed data on the mobile phone ownership of people living in 33 of the 54 countries in Africa. They used mathematical models and data collected from 44,224 people in Afrobarometer, a continent-wide survey conducted between 2017 and 2018. Okano et al. estimated that 80% of African adults in these 33 countries owned a mobile phone, and half of these devices were smartphones.

Although ownership levels varied between the 33 countries, there were substantial inequities that appeared across all of them. More men than women owned a mobile phone. Residents in urban areas and wealthy individuals were also more likely to have a mobile phone than people living in rural areas and poorer individuals, respectively. However, in some countries, the least wealthy were also found to sometimes own smartphones.

Okano et al. also found that people living closer to a health clinic were more likely to have a mobile phone than those living further away. Mobile phone ownership was also higher between 26 to 40 year olds, and then decreased with age. In addition, people under 30 were more likely to have a smartphone, whereas older individuals were more likely to own a mobile phone that does not connect to the internet.

These findings suggest that there are large inequities in mobile phone ownership. If these are not addressed, rolling out mHealth interventions could worsen existing health disparities in African countries. Efforts need to be made across the continent to expand access to phone devices and reduce substantial internet costs. This will ensure that mHealth interventions benefit everyone across Africa, particularly those who need them most.

resources. Such interventions, if shown to be cost-effective, would be extremely important in Africa where many people need to travel long distances to reach healthcare facilities (*World Health Organization, 2021a*). However, in order to use mHealth interventions, individuals need to own a mobile phone; furthermore, many proposed interventions require ownership of smartphones (SPs). In the continent of Africa, mobile phone ownership has been reported to have increased rapidly over the past decade (*Krönke, 2020*), but country-level differences in ownership have neither been quantified nor have country-level inequities in ownership been compared. Here, we address these knowledge gaps for 33, out of the 54, countries in Africa: together, these 33 countries encompass ~60% of the population of the African continent, which contains ~1.4 billion people. Specifically, we: (i) estimate the percentage of individuals who own mobile phones in all 33 countries together, and for each country at the national and sub-national level, (ii) identify a relationship between ownership of a mobile phone and proximity to a health clinic (HC), and (iii) identify inequalities/inequities in the ownership of mobile phones based on gender, urban-rural residency, age, and poverty. We investigate the ownership of a basic mobile phone (BP; a mobile phone that cannot connect to the internet) and a SP (a mobile phone that can connect to the internet) separately. Finally, we discuss the implications of our results for designing and implementing mHealth interventions in Africa.

To conduct our analyses, we used nationally representative sample data collected from 44,224 individuals in 33 countries in Round 7 (R7) of the Afrobarometer survey (*Afrobarometer, 2021*): R7 was conducted in 2017–2018. Afrobarometer is a pan-African, non-partisan research project that has been operating since 1999. It is the world's leading source of high-quality public opinion data for

Africa. The surveys measure citizen's attitudes on democracy, governance, society, the economy, and the continent's progress toward achieving the UN Sustainable Development Goals (**Brass et al., 2019**; **Krönke et al., 2022**; **Mattes, 2019**). The surveys also collect data on the ownership of mobile phones. Further details on the Afrobarometer data are given in the Methods.

Problems due to the geographic inaccessibility of healthcare in Africa have been well documented (**World Health Organization, 2021a**). These problems reflect the resource constraints on the healthcare system, and hence are more acute in some countries in Africa than others, e.g., Botswana is a middle-income country with a well-financed healthcare system, and Malawi is one of the poorest countries in the world with a healthcare system that is severely financially constrained. However, in all African countries, the problem of geographic inaccessibility to healthcare is particularly pronounced in rural areas, where many of the poorest citizens live. In rural areas, the problem of the need to travel long distances to reach HCs is exacerbated by the lack of transportation; many individuals in rural areas have to walk to reach HCs (**Palk et al., 2020**; **World Health Organization, 2021a**). The phenomenon of distance decay in utilization of HCs has been observed in many African countries: the further individuals live from an HC, the less likely they are to utilize healthcare (**Lankowski et al., 2014**). The geographic inaccessibility of HCs has been shown to be associated with decreased utilization of antenatal care and bed nets (for protection against malaria), lower vaccination rates, higher attrition rates from HIV and tuberculosis treatment programs, lower adherence levels in HIV programs, reduced maternal fever-seeking behavior, and uptake in contraception (**Alegana et al., 2018**; **Bilinski et al., 2017**; **Ouma et al., 2017**; **Terzian et al., 2018**).

Currently, there are multiple mHealth interventions that are being investigated or being used at a small scale in almost every African country (**Holst et al., 2020**; **Manby et al., 2022**; **Onukwugha et al., 2022**; **Osei et al., 2021**; **Wood et al., 2019**); the number of mHealth interventions is continuing to increase. Throughout the continent, mHealth interventions are being used for multiple reasons: for disease diagnosis and treatment support by health workers (**Anstey Watkins et al., 2018**; **Wood et al., 2019**), to increase adolescents' use of sexual and reproductive health services (**Onukwugha et al., 2022**), for HIV prevention and management (**Manby et al., 2022**), to improve maternal and child health (**Ag Ahmed et al., 2017**), to support the COVID-19 response (**Fischer et al., 2021**), and to improve surveillance coverage for new outbreaks of infectious diseases (e.g. Ebola; **Tom-Aba et al., 2018**).

## Methods

The Afrobarometer R7 survey (**Afrobarometer, 2021**) collected data in 34 African countries: Benin, Botswana, Burkina Faso, Cabo Verde, Cameroon, Cote d`Ivoire, Eswatini, Gabon, Gambia, Ghana, Guinea, Kenya, Lesotho, Liberia, Madagascar, Malawi, Mali, Mauritius, Morocco, Mozambique, Namibia, Niger, Nigeria, Sao Tome and Principe, Senegal, Sierra Leone, South Africa, Sudan, Tanzania, Togo, Tunisia, Uganda, Zambia, and Zimbabwe. We excluded Kenya from our analysis as its questionnaire did not differentiate between SP and BP ownership.

We used the Afrobarometer data (**Afrobarometer, 2021**) to estimate the probability of owning a mobile phone (either a BP or an SP) and, amongst mobile phone owners, the probability of owning an SP. We made these estimates at three levels: (i) multi-country (aggregating data from all 33 countries), (ii) the national level for each country, and (iii) the sub-national level within each country. To make these estimates, we used data from the n=44,224 individuals in the 33 countries who provided data on mobile phone ownership, i.e., on whether they owned one (n=35,685), did not own one (n=4,903), or did not own one but someone in their house owned one (n=3,383). Individuals who answered 'do not know' or refused to answer were excluded (n=253; proportion of data, $\hat{p}$=0.006). Mobile phone ownership was recoded as a binary variable; participants who reported that someone else in their household owned a mobile phone were coded as not owning a phone. Contingent on mobile phone ownership, participants were asked whether or not their phone had internet access; we assumed that individuals who answered in the affirmative owned an SP (n=16,830), individuals who answered 'do not know' or refused to answer were excluded (n=252; $\hat{p}$=0.007), and we assumed that each of the remaining participants owned a BP (n=18,603). A flow diagram depicting participant sample sizes is provided in **Figure 1—figure supplement 1**. Afrobarometer 'within-country' weights were used for all national-level estimates (**The Afrobarometer Network, 2021**); multi-country estimates were made by weighting the national-level estimates with UN population data (**United Nations, 2019**).

Sub-national estimates of mobile phone ownership were mapped by linking current national and sub-national boundaries (*GADM, 2021*) to Afrobarometer data at the province/state level.

Due to the high cost of internet in African countries, some individuals may own SPs but not pay to access the internet. Therefore, to estimate the percentage of SP owners who may not pay to access the internet, we looked at the frequency of access to the internet within this sub-group. In the Afrobarometer surveys, participants were asked how often they accessed the internet; they were not asked to specify how they accessed the internet. We analyzed these data, stratified on the basis of the type of mobile phone that we assumed individuals owned.

To identify inequalities/inequities in ownership, we used data on five variables from the Afrobarometer R7 survey: gender, age, poverty, urban/rural residency, and proximity to an HC (*Afrobarometer, 2021*). We defined poverty, as in the Afrobarometer surveys, by using the Lived Poverty Index (LPI). This index is calculated by combining answers to five survey questions that measure how often individuals have gone without basic necessities such as water, food, and medical care in the past month (*Mattes, 2020*). We use a four-point scale for LPI, where 0 indicates an individual is in the wealthiest group in terms of accessing basic necessities, and 3 indicates an individual is in the poorest group. Individuals responding to any of the five questions with 'Do not know' were excluded from the analysis (n=465; $\hat{p}$=0.011). We defined proximity to an HC as a binary variable: close, or not. Individuals who had an HC present in the enumeration area of their residence, or within easy walking distance thereof, were considered to be in close proximity to an HC. There were no missing values for proximity to HC or urban/rural residence which were recorded by the interviewer (n=0; $\hat{p}$=0). Individuals who did not know or refused to provide their age (n=39; $\hat{p}$=0.001) or gender (n=7; $\hat{p}$=0.000) were excluded from the analysis. We estimated country-specific crude ORs (cORs) of mobile phone ownership separately by gender and urban-rural status, calculated an age- and gender-stratified population-pyramid of mobile phone ownership, and constructed Bayesian models for (i) the ownership of mobile phones and (ii) the ownership of SPs amongst mobile phone owners.

To specify the Bayesian logistic regression (BLR) models for the probability of owning a mobile phone, we modeled phone ownership $y_{ij}$ of individual $i \in (1, n_j)$ in country $j \in (1, 33)$ as a Bernoulli variable with probability $\theta_{ij}$ :

$$y_{ij}|\theta_{ij} \sim Bern(\theta_{ij}) \tag{1}$$

The probability of owning a mobile phone, $P(y_{ij} = 1) = \theta_{ij}$ , was then modeled using the logit-link function:

$$logit(\theta_{ij}) = \beta_0 + \beta_1 x_{1ij} \tag{2}$$

where $\beta_0$ is the population-level intercept, and $\beta_1$ is a regression coefficient that quantifies the influence of predictor variable $x_{1ij}$ ; we used a separate BLR model for each of the five variables. Notably, $logit(\theta_{ij}) = \ln(\frac{\theta_{ij}}{1-\theta_{ij}})$ is the log-odds of mobile phone ownership. The BLR models for the probability, for mobile phone owners, of owning an SP are defined equivalently, with $y_{ij}$ now representing ownership of an SP by phone owner $i$ in country $j$.

We then constructed Bayesian multilevel logistic regression (BMLR) models (*Gelman et al., 2013*) for the probability of owning a mobile phone (model 1) and the probability, for mobile phone owners, of owning an SP (model 2). These models enabled us to quantify the effect of each of the five variables whilst accounting for the effect of the other four variables and the nested structure of the data. We specified model 1 by modifying *Equation 2*:

$$logit(\theta_{ij}) = \beta_0 + \beta_1 x_{1ij} + \ldots + \beta_5 x_{5ij} + u_{0j} + u_{1j} x_{1ij} + u_{2j} x_{2ij} \tag{3}$$

Here, the five predictor variables are denoted by $x_{kij}$ and their associated regression coefficients by $\beta_k$ ($k \in (1, 5)$). $u_{0j}$ are country-level intercepts, $u_{1j}$ are coefficients for the country-level effect of urban/rural residency $x_{1ij}$ , and $u_{2j}$ are coefficients for the country-level effect of gender $x_{2ij}$ . The country-level effects are distributed as multivariate normal with mean 0 and unstructured covariance matrix $\Sigma$ :

$$\begin{bmatrix} u_{0j} \\ u_{1j} \\ u_{2j} \end{bmatrix} \sim MVN \left( \begin{bmatrix} 0 \\ 0 \\ 0 \end{bmatrix}, \Sigma = \begin{bmatrix} \sigma_{u_0}^2 & \sigma_{u_{01}} & \sigma_{u_{02}} \\ \sigma_{u_{01}} & \sigma_{u_1}^2 & \sigma_{u_{12}} \\ \sigma_{u_{02}} & \sigma_{u_{12}} & \sigma_{u_2}^2 \end{bmatrix} \right) \tag{4}$$

By substituting $\rho_{ab} = \frac{\sigma_{u_{ab}}}{\sigma_{u_a}\sigma_{u_b}}$ (where $\rho_{ab}$ is the correlation between country-level effects $u_a$ and $u_b$), $\Sigma$ can be reparametrized as a function of the correlation matrix ($\boldsymbol{R}$). We derive this for model 1; the result is generalizable.

$$
\begin{aligned}
\Sigma &= \begin{bmatrix} \sigma_{u_0}^2 & \sigma_{u_{01}} & \sigma_{u_{02}} \\ \sigma_{u_{01}} & \sigma_{u_1}^2 & \sigma_{u_{12}} \\ \sigma_{u_{02}} & \sigma_{u_{12}} & \sigma_{u_2}^2 \end{bmatrix} = \begin{bmatrix} \sigma_{u_0}^2 & \rho_{01}\sigma_{u_0}\sigma_{u_1} & \rho_{02}\sigma_{u_0}\sigma_{u_2} \\ \rho_{01}\sigma_{u_0}\sigma_{u_1} & \sigma_{u_1}^2 & \rho_{12}\sigma_{u_1}\sigma_{u_2} \\ \rho_{02}\sigma_{u_0}\sigma_{u_2} & \rho_{12}\sigma_{u_1}\sigma_{u_2} & \sigma_{u_2}^2 \end{bmatrix} \\[2mm]
&= \begin{bmatrix} \sigma_{u_0} & 0 & 0 \\ 0 & \sigma_{u_1} & 0 \\ 0 & 0 & \sigma_{u_2} \end{bmatrix} \underbrace{\begin{bmatrix} 1 & \rho_{01} & \rho_{02} \\ \rho_{01} & 1 & \rho_{12} \\ \rho_{02} & \rho_{12} & 1 \end{bmatrix}}_{\boldsymbol{R}} \begin{bmatrix} \sigma_{u_0} & 0 & 0 \\ 0 & \sigma_{u_1} & 0 \\ 0 & 0 & \sigma_{u_2} \end{bmatrix}
\end{aligned}
\tag{5}
$$

Model 2 for SP ownership (amongst mobile phone owners) is defined in an equivalent manner, with the addition of interaction effects between gender and proximity to an HC. This allows us to discern differences in SP ownership between women who do not live in close proximity to an HC and: (i) women who live in close proximity to an HC, (ii) men who do not live in close proximity to an HC, and (iii) men who live in close proximity to an HC.

We used weakly informative priors (*Gelman, 2006*; *Gelman et al., 2013*; *Gelman et al., 2008*; *Lemoine, 2019*; *Williams et al., 2018*):

$$
\begin{aligned}
\beta_0 &\sim N\left(0, 5\right) \\
\beta_k &\sim N\left(0, 1.5\right) \\
\sigma_{u_l} &\sim \text{Half-}t\left(0, 2.5\right) \\
\boldsymbol{R} &\sim LKJ\left(2\right)
\end{aligned}
\tag{6}
$$

For the population-level effects $\beta_k(k \in (1, N))$, we chose normal priors with mean 0 and SD 1.5. This necessitates the initial sampling to not exclude effects in excess of ±3.9, i.e., the 99% boundary for this distribution. This corresponds to being able to detect an OR as large as 48, or as small as 1/48. We used similar reasoning to define the other priors. We used half-t priors for the SD parameters (*Gelman, 2006*) to ensure positivity. Finally, we used an LKJ(2) prior for the correlation matrix $\boldsymbol{R}$; this distribution (*Lewandowski et al., 2009*) is bounded by [–1, 1] and centered at 0 with edge values less likely.

We fit the BLR and BMLR models by using Markov Chain Monte Carlo (MCMC) sampling to approximate the posterior distributions of all model parameters. MCMC sampling was conducted in the programming language Stan (*Stan Development Team, 2020*) via the R package brms (v. 2.15.0; *Bürkner, 2017*). 10,000 posterior samples were drawn over 4 chains, following a warm-up of 1,000 for each. Standard MCMC diagnostics (*Betancourt, 2017*; *Gelman et al., 2013*; *Gelman and Rubin, 1992*) were used to assess chain convergence, independence, and sampling efficiency. We made density plots of each parameter's posterior distribution: medians and Bayesian 95% highest posterior density (HPD) regions. For all five variables, we computed median ORs (cORs for the BLR models, and aORs for the BMLR models) and the corresponding 95% HPD regions. ROC curves were used to assess the diagnostic capabilities of the fitted BMLR models.

## Results

We found that 82% of individuals in the 33 countries own mobile phones; 42% of individuals own BPs and 40% own SPs. Notably, we found that only 13% of individuals that we classified as SP owners (and 89% of individuals that we classified as owners of BP) reported that they never accessed the internet (*Figure 1—figure supplement 2*). *Figure 1A* shows the probability of phone ownership in all

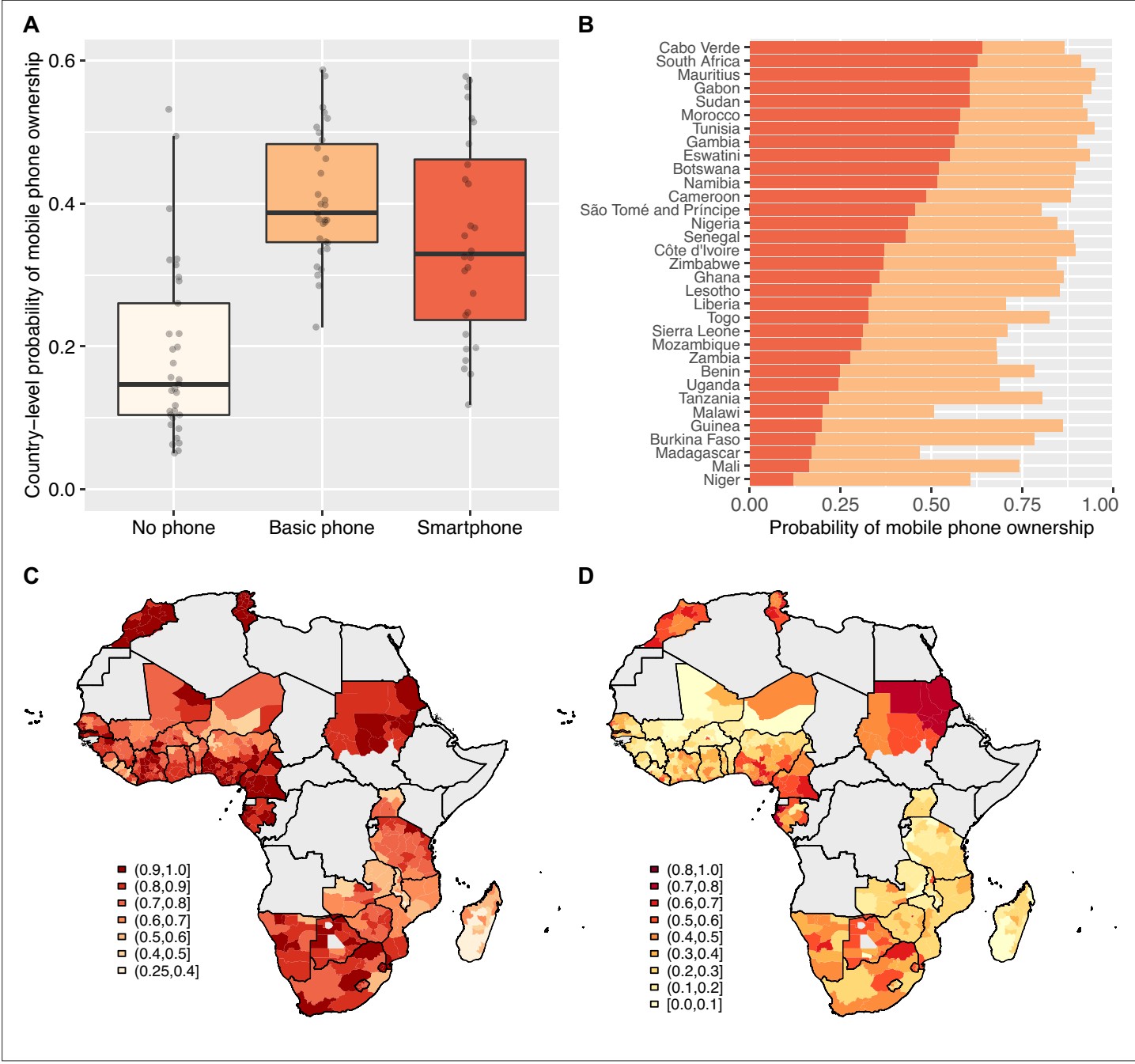

**Figure 1.** Basic mobile phone and smartphone ownership in 33 African countries. (**A**) Boxplots show the probabilities of not owning a mobile phone (cream), owning a basic mobile phone (BP; orange), or owning a smartphone (SP; red). Country-level probabilities (dots) are overlaid and jittered to reduce overlap. (**B**) Barplot shows the country-level probabilities of BP ownership (orange) and SP ownership (red) ordered by SP ownership. Geographic distribution showing probabilities of (**C**) BP ownership and (**D**) SP ownership in 33 Afrobarometer countries at the sub-national level.

The online version of this article includes the following figure supplement(s) for figure 1:

**Figure supplement 1.** Flow diagram of participant sample sizes.

**Figure supplement 2.** Internet usage by type of phone owned.

**Figure supplement 3.** Phone ownership in the poorest individuals.

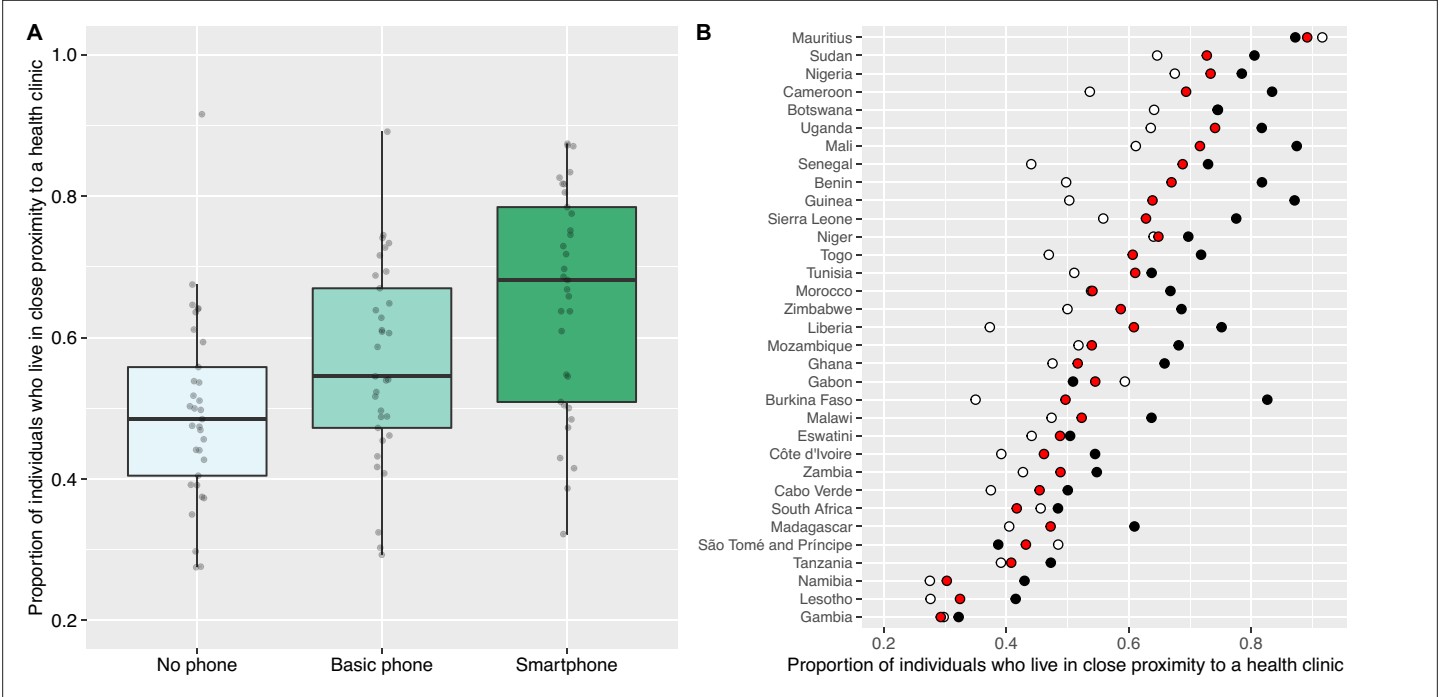

**Figure 2.** Proximity to health clinics and mobile phone ownership in 33 African countries. (**A**) Boxplots show the proportion of individuals who live in close proximity to a health clinic (HC) based on whether they do not own a mobile phone (mean 0.49), own a basic mobile phone (BP; mean 0.56), or own a smartphone (SP; mean 0.66). Country-level probabilities (dots) are overlaid and jittered to reduce overlap. (**B**) Scatterplot shows the country-specific proportions of individuals who live in close proximity to an HC amongst individuals who: do not own a mobile phone (white dots), own a BP (red dots), or own an SP (black dots).

33 countries. Overall, the probability of not owning a mobile phone (either a BP or an SP) is fairly low (median: 0.15), but there is substantial variation between countries: the probability ranges from 0.05 to 0.53 with SD 0.12. The probability of owning a BP is moderately high (median: 0.40), but again – there is substantial variation amongst countries, ranging from 0.23 to 0.66 with SD 0.10. Overall, the country-level probability of owning a BP is not significantly different (t=0.76; p=0.45) from the probability of owning an SP, but the probability of owning an SP is much more variable amongst countries (SD=0.16). *Figure 1B* shows the country-specific probabilities of ownership of a BP (orange data) or an SP (red data). Notably, there is substantial sub-national variation, within almost every one of the 33 countries, in the ownership of a BP (*Figure 1C*) or of an SP (*Figure 1D*). Even in countries such as South Africa, where there is a high probability of owning a mobile phone, there is a very low probability of ownership in certain areas of the country.

We found a very clear relationship between country-level ownership of a mobile phone and living in close proximity to an HC (*Figure 2A*). SP owners were more likely than BP owners to live close to an HC (t=6.86; p<0.001); in turn, BP owners were significantly more likely than individuals who did not own a mobile phone to live close to an HC (t=5.52; p<0.001). *Figure 2B* shows, for each of the 33 countries, the proportion of individuals who live in close proximity to an HC based on whether they own an SP, own a BP, or do not own a mobile phone. Again, there is considerable variation amongst countries.

Considering the 33 countries together, we found a gender inequity in ownership of mobile phones: 87% of men versus 76% of women own mobile phones. We also found a gender inequity in ownership of SPs: 50% of men (who own mobile phones) own SPs versus 46% of women (who own mobile phones). We found that men are significantly more likely than women to own mobile phones in 27 out of the 33 countries (*Table 1*): the OR of male to female phone ownership is greatest in Benin (cOR 4.78). We did not find a significant gender difference in Botswana, Cabo Verde, Gabon, Lesotho, Namibia, or South Africa. Amongst mobile phone owners, men are significantly more likely than women to own SPs in 15 of the 33 countries (*Table 1*): gender inequity is most pronounced in Benin (cOR 2.84).

**Table 1.** Probability of mobile phone ownership by country and gender.

| Country | Any mobile phone | | | Smartphone | | |
|---|---|---|---|---|---|---|
| | Female | Male | OR[†] | Female | Male | OR[†] |
| Benin | 0.66 | 0.90 | 4.78* | 0.19 | 0.41 | 2.84* |
| Mali | 0.60 | 0.88 | 4.77* | 0.15 | 0.26 | 1.93* |
| Burkina Faso | 0.67 | 0.90 | 4.31* | 0.18 | 0.27 | 1.64* |
| Senegal | 0.83 | 0.95 | 3.90* | 0.44 | 0.52 | 1.40* |
| Guinea | 0.79 | 0.93 | 3.46* | 0.22 | 0.23 | 1.06 |
| Niger | 0.47 | 0.74 | 3.23* | 0.14 | 0.23 | 1.92* |
| Cote d`Ivoire | 0.85 | 0.95 | 3.10* | 0.32 | 0.49 | 1.98* |
| Nigeria | 0.78 | 0.91 | 3.02* | 0.48 | 0.54 | 1.29* |
| Gambia | 0.85 | 0.94 | 2.82* | 0.60 | 0.65 | 1.23 |
| Togo | 0.75 | 0.89 | 2.80* | 0.33 | 0.45 | 1.61* |
| Uganda | 0.59 | 0.80 | 2.76* | 0.33 | 0.37 | 1.18 |
| Tunisia | 0.92 | 0.97 | 2.45* | 0.61 | 0.60 | 0.97 |
| Sierra Leone | 0.62 | 0.80 | 2.42* | 0.44 | 0.44 | 1.01 |
| Tanzania | 0.74 | 0.87 | 2.41* | 0.23 | 0.31 | 1.50* |
| Malawi | 0.41 | 0.61 | 2.29* | 0.34 | 0.43 | 1.49* |
| Morocco | 0.90 | 0.95 | 2.28* | 0.60 | 0.64 | 1.22 |
| Ghana | 0.82 | 0.91 | 2.15* | 0.35 | 0.47 | 1.70* |
| Mauritius | 0.93 | 0.97 | 2.11* | 0.63 | 0.64 | 1.02 |
| Cameroon | 0.85 | 0.92 | 2.10* | 0.55 | 0.54 | 0.96 |
| Mozambique | 0.61 | 0.75 | 1.87* | 0.45 | 0.45 | 1.02 |
| Sao Tome and Principe | 0.75 | 0.85 | 1.83* | 0.55 | 0.58 | 1.14 |
| Liberia | 0.64 | 0.76 | 1.79* | 0.38 | 0.53 | 1.90* |
| Sudan | 0.89 | 0.94 | 1.77* | 0.65 | 0.67 | 1.11 |
| Eswatini | 0.92 | 0.95 | 1.74* | 0.57 | 0.60 | 1.14 |
| Zambia | 0.63 | 0.73 | 1.61* | 0.40 | 0.41 | 1.06 |
| Zimbabwe | 0.82 | 0.87 | 1.46* | 0.42 | 0.45 | 1.15 |
| Madagascar | 0.44 | 0.50 | 1.29* | 0.36 | 0.36 | 1.00 |
| Gabon | 0.93 | 0.94 | 1.26 | 0.63 | 0.66 | 1.13 |
| Namibia | 0.88 | 0.90 | 1.24 | 0.54 | 0.62 | 1.40* |
| Botswana | 0.89 | 0.91 | 1.23 | 0.55 | 0.62 | 1.33* |
| Cabo Verde | 0.85 | 0.88 | 1.20 | 0.71 | 0.77 | 1.36* |
| Lesotho | 0.84 | 0.86 | 1.18 | 0.39 | 0.39 | 1.02 |
| South Africa | 0.91 | 0.91 | 0.97 | 0.69 | 0.68 | 0.93 |

*Significant at α=0.05.
†Odds ratio of male phone ownership to female phone ownership.

Considering the 33 countries together, we found a substantial inequity in ownership of mobile phones based on whether individuals lived in urban or rural areas: 91% of urban residents versus 74% of rural residents. We also found an urban-rural inequity in ownership of SPs: 61% of urban residents (who own mobile phones) own SPs versus 35% of rural residents (who own mobile phones). We

**Table 2.** Probability of mobile phone ownership by country and urban/rural status.

| Country | Any mobile phone | | | Smartphone | | |
|---|---|---|---|---|---|---|
| | Rural | Urban | OR† | Rural | Urban | OR† |
| Gabon | 0.82 | 0.97 | 7.17* | 0.31 | 0.72 | 5.66* |
| Zimbabwe | 0.78 | 0.95 | 5.94* | 0.28 | 0.64 | 4.58* |
| Uganda | 0.63 | 0.90 | 5.58* | 0.29 | 0.49 | 2.42* |
| Burkina Faso | 0.74 | 0.94 | 5.25* | 0.12 | 0.52 | 7.57* |
| Ghana | 0.77 | 0.94 | 5.09* | 0.24 | 0.53 | 3.60* |
| Senegal | 0.82 | 0.96 | 4.78* | 0.34 | 0.60 | 2.90* |
| Morocco | 0.86 | 0.97 | 4.70* | 0.42 | 0.71 | 3.51* |
| Zambia | 0.56 | 0.84 | 4.23* | 0.29 | 0.50 | 2.54* |
| Madagascar | 0.40 | 0.73 | 4.17* | 0.27 | 0.54 | 3.25* |
| Togo | 0.76 | 0.93 | 4.14* | 0.25 | 0.57 | 3.92* |
| Tanzania | 0.75 | 0.92 | 4.00* | 0.17 | 0.42 | 3.45* |
| Malawi | 0.45 | 0.76 | 3.87* | 0.33 | 0.54 | 2.37* |
| Guinea | 0.82 | 0.95 | 3.84* | 0.12 | 0.42 | 5.11* |
| Liberia | 0.58 | 0.84 | 3.69* | 0.35 | 0.55 | 2.33* |
| Botswana | 0.84 | 0.95 | 3.65* | 0.43 | 0.73 | 3.56* |
| Mali | 0.70 | 0.88 | 3.24* | 0.13 | 0.47 | 6.09* |
| Niger | 0.57 | 0.80 | 3.10* | 0.14 | 0.37 | 3.63* |
| Sudan | 0.89 | 0.96 | 3.09* | 0.59 | 0.76 | 2.16* |
| Nigeria | 0.79 | 0.92 | 3.05* | 0.42 | 0.61 | 2.20* |
| Lesotho | 0.82 | 0.93 | 2.91* | 0.27 | 0.57 | 3.62* |
| Mozambique | 0.60 | 0.81 | 2.86* | 0.33 | 0.60 | 3.02* |
| Cameroon | 0.83 | 0.93 | 2.74* | 0.43 | 0.64 | 2.42* |
| Cote d`Ivoire | 0.86 | 0.94 | 2.45* | 0.26 | 0.55 | 3.35* |
| Tunisia | 0.91 | 0.96 | 2.32* | 0.52 | 0.64 | 1.62* |
| Sierra Leone | 0.64 | 0.80 | 2.28* | 0.32 | 0.55 | 2.61* |
| Namibia | 0.85 | 0.93 | 2.22* | 0.44 | 0.68 | 2.72* |
| Benin | 0.72 | 0.85 | 2.18* | 0.23 | 0.40 | 2.21* |
| Eswatini | 0.93 | 0.97 | 2.17 | 0.56 | 0.73 | 2.17* |
| Cabo Verde | 0.82 | 0.89 | 1.69* | 0.63 | 0.79 | 2.26* |
| South Africa | 0.89 | 0.92 | 1.36 | 0.57 | 0.74 | 2.07* |
| Sao Tome and Principe | 0.77 | 0.82 | 1.34* | 0.52 | 0.59 | 1.30 |
| Mauritius | 0.95 | 0.95 | 0.98 | 0.61 | 0.67 | 1.29* |
| Gambia | 0.92 | 0.89 | 0.71 | 0.61 | 0.64 | 1.15 |

*Significant at α=0.05.
†Odds ratio of urban phone ownership to rural phone ownership.

found significant urban-rural differences in the ownership of mobile phones in 29 of the 33 countries (*Table 2*); the OR of urban to rural phone ownership is greatest in Gabon (cOR 7.17). Notably, we did not find a significant urban-rural inequity in Eswatini, the Gambia, Mauritius, or South Africa. Urban residents are more likely to own SPs than rural residents, in 31 of 33 countries (*Table 2*); the urban/

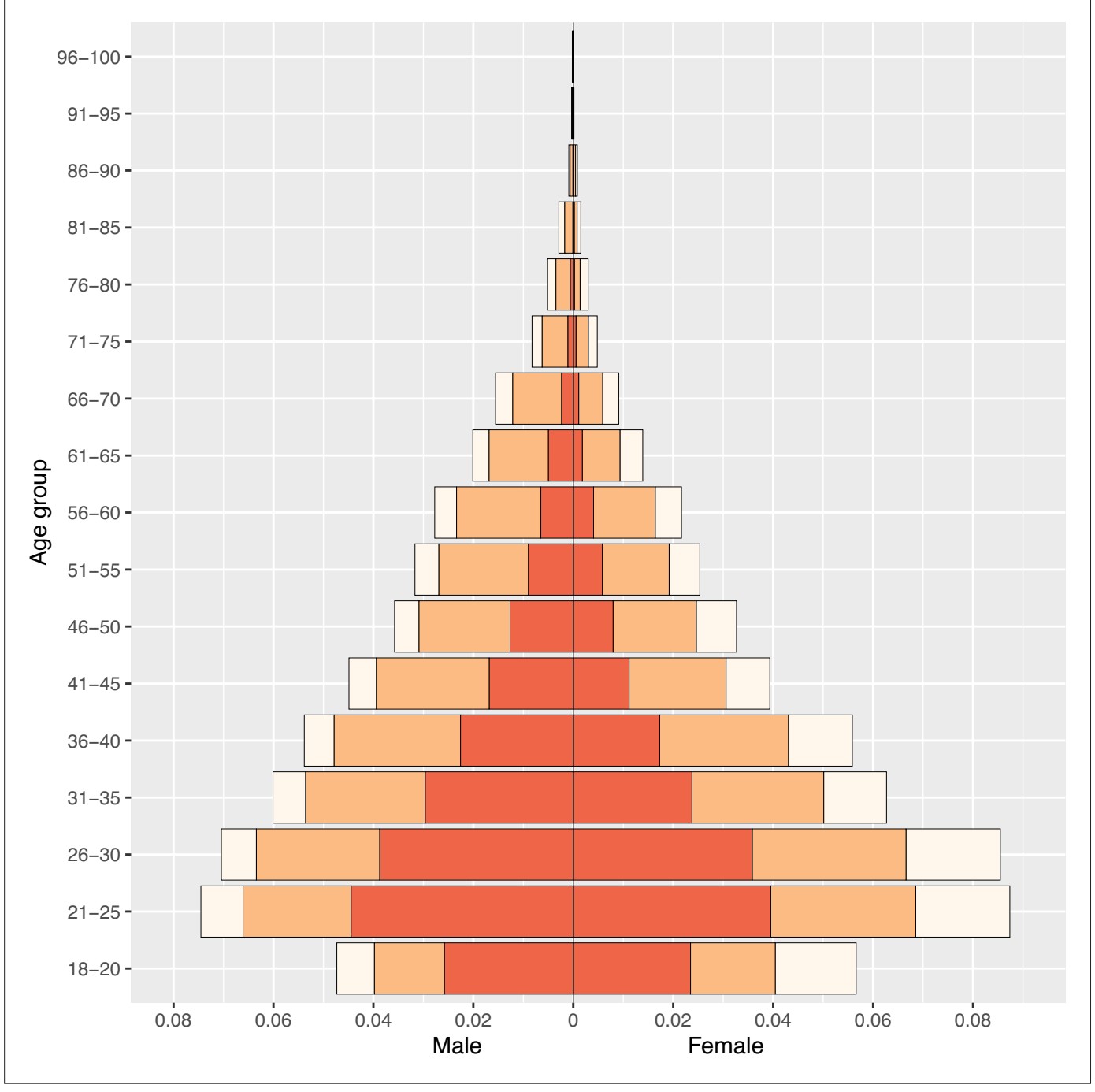

**Figure 3.** Phone ownership by age and gender in 33 African countries. Population pyramid displays the distribution of the population stratified by gender and 5 year age groupings (with the exception of the 18–20 age class) by ownership of a mobile phone: no mobile phone (cream), a basic mobile phone (BP; orange), or a smartphone (SP; red).

rural difference in SP ownership is greatest in Burkina Faso (cOR 7.57). We did not find significant urban-rural differences in ownership of SPs in the Gambia or Sao Tome and Principe.

The age and gender-stratified ownership pyramid (*Figure 3*) is based upon aggregated data from all 33 countries. The age pyramid shows that ownership of mobile phones differs substantially amongst age classes, by type of mobile phone (BP or SP) and by gender. In all age classes, a high proportion of individuals own mobile phones; almost half of phone owners are under 30 years old. For women, 18–30 year olds (and for men, 18–35 year olds) are more likely to own an SP than a BP; whereas,

**Table 3.** Fitted Bayesian models for determinants of mobile phone ownership in Africa.

| Population-level effects | Bivariable (Bayesian logistic regression [BLR] models) | | Multivariable (Bayesian multilevel logistic regression [BMLR] model 1) | |
| --- | --- | --- | --- | --- |
| | Crude OR | 95% highest posterior density (HPD) region[†] | aOR | 95% highest posterior density (HPD) region[†] |
| Female | ref | - | ref | - |
| Male | 2.04 | (1.94, 2.16)* | 2.37 | (1.96, 2.84)* |
| Rural | ref | - | ref | - |
| Urban | 3.47 | (3.27, 3.68)* | 2.66 | (2.22, 3.18)* |
| Age (61+) | ref | - | ref | - |
| 18–20 | 1.36 | (1.21, 1.53)* | 1.73 | (1.51, 1.96)* |
| 21–25 | 1.75 | (1.59, 1.93)* | 2.29 | (2.05, 2.56)* |
| 26–40 | 1.98 | (1.82, 2.15)* | 2.68 | (2.44, 2.94)* |
| 41–50 | 1.85 | (1.68, 2.03)* | 2.44 | (2.19, 2.72)* |
| 51–60 | 1.54 | (1.39, 1.71)* | 1.78 | (1.59, 2.00)* |
| Lived Poverty Index (LPI) = 0 (wealthiest) | ref | - | ref | - |
| LPI = 1 | 0.50 | (0.44, 0.56)* | 0.69 | (0.61, 0.78)* |
| LPI = 2 | 0.29 | (0.26, 0.32)* | 0.46 | (0.41, 0.52)* |
| LPI = 3 (poorest) | 0.24 | (0.21, 0.26)* | 0.35 | (0.31, 0.40)* |
| No health clinic (HC) in close proximity | ref | - | ref | - |
| HC in close proximity | 1.56 | (1.49, 1.65)* | 1.31 | (1.24, 1.39)* |

| Country-level effects | | | Estimate | 95% highest posterior density (HPD) region[†] |
| --- | --- | --- | --- | --- |
| $\sigma_{u_0}$ | | | 0.87 | (0.67, 1.11) |
| $\sigma_{u_1}$ | | | 0.48 | (0.33, 0.62) |
| $\sigma_{u_2}$ | | | 0.49 | (0.36, 0.64) |
| $\rho_{01}$ | | | –0.43 | (–0.73, –0.13) |
| $\rho_{02}$ | | | –0.32 | (–0.63, –0.02) |
| $\rho_{12}$ | | | 0.21 | (–0.16, 0.55) |

*Significant at α=0.05.
[†]Bayesian 95% HPD regions.

the opposite holds true in the older age classes. In all age classes, a greater proportion of men than women own mobile phones; this gender inequity in ownership is accentuated for SPs.

Results from the BLR and BMLR models on the probability of owning a mobile phone are shown in *Table 3*. All ORs are significantly different from one in both the bivariable and multivariable analysis. Men have over twice the odds of owning a mobile phone than women (aOR: 2.37, 95% HPD region: 1.96–2.84). Urban residents have nearly three times the odds of owning a mobile phone than rural residents (aOR: 2.66, 95% HPD region: 2.22–3.18). Ownership of mobile phones increases with age, peaks in 26–40 year olds, and then decreases. The probability of ownership decreases with the LPI; the wealthiest individuals have approximately three times higher odds of owning mobile phones than the poorest individuals (aOR: 2.87, 95% HPD region: 2.53–3.27). Notably, individuals who live in close proximity to an HC have higher odds of owning mobile phones than individuals who do not live in close proximity to an HC (aOR: 1.31, 95% HPD region: 1.24–1.39). Country-specific effects are apparent for the inequity in ownership based on gender (*Figure 4A*) and urban/rural residency (*Figure 4B*), and for

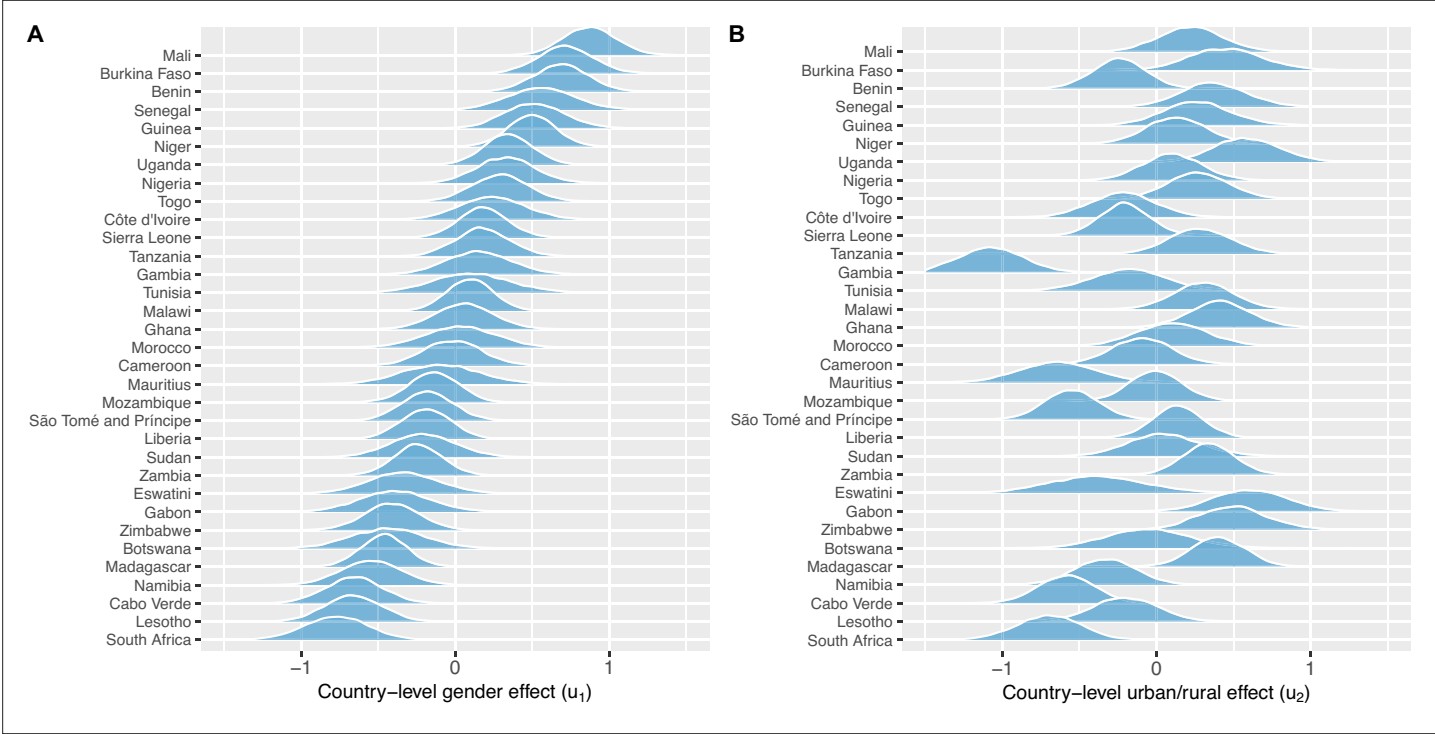

**Figure 4.** Country-level gender and urban-rural effects. (**A**) Posterior distributions of the country-level effect on mobile phone ownership of being male (compared to female), sorted by median. (**B**) Posterior distributions of the country-level effect of living in an urban area (compared to living in a rural area), in the same country order as (**A**). Both (**A**) and (**B**) are on the logit-scale – and should be viewed respectively as country-specific adjustments to the population-level effect of (**A**) being male or (**B**) living in an urban area.

The online version of this article includes the following figure supplement(s) for figure 4:

**Figure supplement 1.** Model 1 posteriors – (country-level) intercept.

**Figure supplement 2.** Model 2 posteriors – (country-level) intercept.

**Figure supplement 3.** Model 2 posteriors – (country-level) urban/rural.

**Figure supplement 4.** Model 2 posteriors – (country-level) gender and proximity I.

**Figure supplement 5.** Model 2 posteriors – (country-level) gender and proximity II.

**Figure supplement 6.** Model 2 posteriors – (country-level) gender and proximity III.

**Figure supplement 7.** Model 1 Markov Chain Monte Carlo (MCMC) diagnostics and ROC curve.

**Figure supplement 8.** Model 2 Markov Chain Monte Carlo (MCMC) diagnostics and ROC curve.

the intercept (*Figure 4—figure supplement 1*), i.e., certain countries have greater (or lesser) inequities than the average effect for the 33 countries.

For mobile phone owners who own an SP, the results from the BLR and BMLR models are shown in *Table 4*. All ORs are significantly different from one in both the bivariable and multivariable analysis. We found an interaction effect between gender and living in proximity to an HC; specifically, the effect of gender on owning an SP depends upon proximity to an HC. Men who do not live in close proximity to an HC had higher odds of SP ownership (aOR: 1.50, 95% HPD region: 1.30–1.72) than women who do not live in close proximity (1; the baseline category). The gender difference is accentuated by proximity to an HC: men in close proximity to an HC had even larger odds of owning an SP (aOR: 1.92, 95% HPD region: 1.63–2.26) than women in close proximity (aOR: 1.15, 95% HPD region: 1.03–1.30). Urban residents (who own mobile phones) are nearly three times as likely as rural residents (who own mobile phones) to own SPs (aOR: 2.67, 95% HPD region: 2.33–3.10). Ownership of SPs (amongst owners of mobile phones) is most likely in 18–30 year olds and decreases with age. The odds of ownership decrease with the LPI: the wealthiest individuals (who own mobile phones) are approximately three times more likely than the poorest individuals (who own mobile phones) to own SPs (aOR: 2.86, 95% HPD region: 2.57–3.17). Country-specific effects are shown in *Figure 4—figure*

**Table 4.** Fitted Bayesian models for determinants of smartphone ownership among mobile phone owners in Africa.

| Population-level effects | Bivariable (Bayesian logistic regression [BLR] models) | | Multivariable (Bayesian multilevel logistic regression [BMLR] model 2) | |
| --- | --- | --- | --- | --- |
| | Crude OR | 95% highest posterior density (HPD) region† | aOR | 95% highest posterior density (HPD) region† |
| Female, no health clinic (HC) in close proximity | ref | - | ref | - |
| Female, HC in close proximity | 1.23 | (1.15, 1.32)* | 1.15 | (1.03, 1.30)* |
| Male, no HC in close proximity | 1.10 | (1.03, 1.18)* | 1.50 | (1.30, 1.72)* |
| Male, HC in close proximity | 1.54 | (1.44, 1.65)* | 1.92 | (1.63, 2.26)* |
| Rural | ref | - | ref | - |
| Urban | 2.95 | (2.81, 3.09)* | 2.67 | (2.33, 3.10)* |
| Age (61+) | ref | - | ref | - |
| 18–30 | 5.59 | (5.08, 6.16)* | 7.85 | (7.04, 8.77)* |
| 31–40 | 3.53 | (3.19, 3.90)* | 4.73 | (4.24, 5.30)* |
| 41–50 | 2.25 | (2.02, 2.50)* | 2.75 | (2.44, 3.09)* |
| 51–60 | 1.58 | (1.41, 1.77)* | 1.72 | (1.51, 1.96)* |
| Lived Poverty Index (LPI) = 0 (wealthiest) | ref | - | ref | - |
| LPI = 1 | 0.61 | (0.57, 0.66)* | 0.67 | (0.62, 0.73)* |
| LPI = 2 | 0.38 | (0.35, 0.40)* | 0.45 | (0.41, 0.49)* |
| LPI = 3 (poorest) | 0.27 | (0.25, 0.29)* | 0.35 | (0.32, 0.39)* |
| **Country-level effects** | | | **Estimate** | **95% highest posterior density (HPD) region†** |
| $\sigma_{u_0}$ | | | 0.85 | (0.67, 1.05) |
| $\sigma_{u_1}$ | | | 0.39 | (0.28, 0.50) |
| $\sigma_{u_2}$ | | | 0.21 | (0.08, 0.34) |
| $\sigma_{u_3}$ | | | 0.31 | (0.18, 0.43) |
| $\sigma_{u_4}$ | | | 0.41 | (0.28, 0.53) |
| $\rho_{01}$ | | | –0.61 | (–0.81, –0.37) |
| $\rho_{02}$ | | | –0.38 | (–0.77, 0.01) |
| $\rho_{03}$ | | | –0.51 | (–0.80, –0.19) |
| $\rho_{04}$ | | | –0.78 | (–0.93, –0.60) |
| $\rho_{12}$ | | | 0.10 | (–0.38, 0.54) |
| $\rho_{13}$ | | | 0.16 | (–0.24, 0.55) |
| $\rho_{14}$ | | | 0.29 | (–0.06, 0.60) |
| $\rho_{23}$ | | | –0.07 | (–0.59, 0.41) |
| $\rho_{24}$ | | | 0.47 | (0.04, 0.82) |
| $\rho_{34}$ | | | 0.61 | (0.30, 0.88) |

*Significant at α=0.05.
†Bayesian 95% HPD regions.

*supplements 2–6*. Notably, in many of the 33 countries, ownership of mobile phones is relatively high even amongst the poorest of the wealth classes; this holds true for both BPs and, more surprisingly, SPs (*Figure 1—figure supplement 3*). MCMC diagnostic plots and ROC curves for models 1 and 2 are provided in *Figure 4—figure supplements 7–8*.

## Discussion

Considering all 33 African countries together, our results show that a fairly high proportion (82%) of individuals own a mobile phone, and ownership of either a BP or an SP is equally likely; however, we found considerable variability – amongst (and within) the 33 countries – in the proportion of individuals who own mobile phones. Furthermore, we found substantial inequities in ownership based on: gender (with men significantly more likely to own mobile phones than women), the urban-rural divide (with urban residents significantly more likely to own mobile phones than rural residents), and the LPI (wealthy individuals significantly more likely to own mobile phones than poor individuals). Surprisingly, we found that some of the poorest individuals in all 33 countries own SPs. All of the inequities were exacerbated when ownership of an SP was considered in comparison to ownership of a BP. We also found that ownership increases with age, peaks in 26–40 year olds, and then decreases. Due to the demographic structure of Africa, the majority of phone owners are under 30 years old. In this age category, individuals are more likely to own an SP than a BP; older individuals are more likely to own a BP than an SP. Notably, we found that not all countries had gender inequities in ownership of mobile phones, but – essentially – all countries have substantial urban-rural inequities in ownership. In future work, we will identify explanatory factors underlying the inequities in phone ownership that we have found exist both between, and within, countries.

There are many potential factors that have been identified in previous studies that may explain the inequalities in ownership of mobile phones that we have found. Women may be less likely than men to own mobile phones due to receiving less education (which results in lower levels of literacy and digital literacy), higher levels of poverty, and cultural norms of patriarchy that prevent female empowerment and reduce the agency of women (*GSMA, 2022*; *LeFevre et al., 2020*; *Marron et al., 2020*; *Paradigm Initiative, 2020*; *Porter et al., 2020*). The rural population may be less likely than urban residents to own mobile phones due to many of the same factors (fewer years of schooling, lower levels of literacy and digital literacy, and higher levels of poverty; *GSMA, 2020*; *Houngbonon et al., 2021*; *Wesolowski et al., 2012*). However, the main driver of the inequality may be that there is less infrastructure in rural areas than in urban areas (*Krönke, 2020*; *Krönke et al., 2022*); rural areas have lower cell coverage, a less reliable source of electricity, and often lack high-speed broadband (*Houngbonon et al., 2021*). Not surprisingly, we found that wealthy individuals are more likely to own mobile phones than less wealthy individuals; this effect was exacerbated when considering SPs relative to BPs. However, in each of the 33 countries that we have investigated, we found that some of the poorest individuals reported owning SPs. This may be explained by the fact that, over the past few years in Africa, SPs have increasingly become an essential tool for improving an individuals' livelihood and daily affairs (*Alliance for Affordable Internet, 2020*; *World Health Organization, 2021b*). SPs are used to access government services and healthcare (e.g. to register for COVID-19 vaccination), for agricultural purposes, commerce, education, expanding employment opportunities, promoting entrepreneurialism, and electronic money transfer (*Alliance for Affordable Internet, 2020*; *Etzo and Collender, 2010*; *Marsh et al., 2020*; *Porter et al., 2018*; *Poushter and Oates, 2015*; *Quandt et al., 2020*; *World Health Organization, 2021b*). Additionally, some of the poorest individuals may own SPs due to their participation in the national Village Phone Program (*Futch and McIntosh, 2009*), which operates in multiple African countries such as Nigeria, Uganda, and Rwanda. An individual subscribes to the program by taking a loan and buying an SP; they are then trained on how to operate the SP and to make a profit by charging others to use it.

Many of the inequalities that we have found exist in the ownership of mobile phones are the same inequalities that exist in accessing healthcare and health-related information. Individuals living in rural areas – due to the geographic distribution of healthcare facilities – have the greatest difficulty in accessing healthcare and health-related information (*Palk et al., 2020*; *World Health Organization, 2021a*) and – as we have shown – are less likely to own mobile phones, particularly SPs, than individuals living in urban areas. Similarly, the most vulnerable members of society (women, children, the elderly, and those living in poverty) have the greatest difficulty in accessing healthcare and health-related

information and – as we have shown – are also less likely to own mobile phones, particularly SPs, than men, young adults, and the wealthy. Notably, these vulnerable members of society are often the most in need of healthcare, e.g., women are in need of healthcare personnel for assistance with childbirth and antenatal care, and children are in need of protection against malaria and vaccination against childhood diseases.

mHealth approaches are currently being used, at a small-scale, to diagnose and treat over a dozen diseases and health conditions in Africa (*Holst et al., 2020*; *Manby et al., 2022*; *Onukwugha et al., 2022*; *Osei et al., 2021*; *Wood et al., 2019*). Treatment of infectious diseases of relatively long duration, such as tuberculosis and HIV, has received the most attention (*Manby et al., 2022*; *Maraba et al., 2018*; *Nwaozuru et al., 2021*; *Osei et al., 2021*). However, mHealth interventions are also being explored for non-communicable diseases, such as diabetes (*Dike et al., 2021*), cancer (*Mutebi et al., 2020*), and conditions such as alcohol consumption (*Suleman et al., 2021*) and hearing loss (*Bhamjee et al., 2022*). Throughout Africa, many pilot mHealth interventions have been launched to manage Ebola (*Tom-Aba et al., 2018*), maternal health (*Ag Ahmed et al., 2017*; *Mildon and Sellen, 2019*; *Onukwugha et al., 2022*), and various childhood ailments (*Mahmood et al., 2020*). To date, the majority of mHealth interventions that have been implemented in Africa have focused on using SMS/texting to improve treatment adherence and patient retention (*Kruse et al., 2019*; *Manby et al., 2022*; *Odukoya et al., 2021*); notably, such interventions only require access to BPs. Studies of these interventions have shown that weekly text messaging to HIV patients that are literate (and voice-enabled systems for illiterate patients) have substantially improved adherence to medication and increased retention in care. mHealth approaches in Africa are currently being investigated for their utility in increasing access to medical education and training for health workers (*Anstey Watkins et al., 2018*), as a platform to support nurses and midwives (*Nigussie et al., 2021*), to improve men's access to HIV self-testing (*van Heerden et al., 2017*), public health messaging (*Downs et al., 2019*), disease surveillance (*Brinkel et al., 2014*; *Wood et al., 2019*), symptom monitoring (*Lai et al., 2019*; *Wood et al., 2019*), epidemic outbreak tracking (*Singh et al., 2019*), and to aid vaccination campaigns (e.g. for COVID-19; *Fischer et al., 2021*; *Oliver-Williams et al., 2017*). Several systematic reviews have concluded that some (but not all) of the mHealth programs that have been initiated have a positive impact on health behaviors and outcomes, especially in rural and remote communities (*Ag Ahmed et al., 2017*; *Mbuthia et al., 2019*; *Mildon and Sellen, 2019*; *Oliver-Williams et al., 2017*).

Our results have important implications for designing and rolling out mHealth interventions in Africa; mHealth interventions have the potential to increase the quality, reduce the cost, and extend the reach of healthcare (*Holst et al., 2020*; *Manby et al., 2022*; *Onukwugha et al., 2022*; *Osei et al., 2020*; *Wood et al., 2019*). Those who are most in need of mHealth interventions are those in rural areas who do not live in close proximity to an HC; these individuals will need to own mobile phones to access these mHealth interventions. However, our results show that these individuals are currently far less likely to own mobile phones than individuals who may have less need of mHealth interventions, i.e., individuals in urban areas who live in close proximity to an HC. Notably, our results demonstrate that mHealth interventions need to be designed to take age, geographic variation, and inequities (based on gender, urban/rural residency and wealth) in ownership of BPs and SPs into consideration. Previous studies have shown that it will also be critical to consider the impact of education and low levels of literacy on the ownership and usage of mobile phones and hence on the design of mHealth interventions (*Krönke, 2020*). Our results suggest that, due to current levels of mobile phone ownership, it may only be possible to scale up mHealth interventions in a few countries (e.g. Botswana; *United Nations, 2021*), but not in the vast majority of the 33 countries that we have analyzed. SPs are not yet widely used in many African countries; hence in the foreseeable future, high-tech mHealth interventions that require SPs (and regular access to the internet), rather than lower-tech interventions that only require BPs, will be hard to implement in the low-ownership countries that we have identified in our analyses. Taken together, our results demonstrate that before the promise of mHealth interventions can be reached in Africa, there is a need for a 'digital transformation' to occur throughout the continent.

In 2019, the United Nations Broadband Commission for Sustainable Development proposed that there was a need for a 'Digital Infrastructure Moonshot for Africa' (*Broadband Commission, 2019*). In 2020, the African Union (a continental body consisting of the member states that make up the countries of the African Continent) outlined specific goals that need to be met. 'By 2030,

all our people should be digitally empowered and able to access safely and securely to at least 6 megabytes per second all the time where ever they live in the continent at an affordable price of no more than 1 cent (US dollars) per megabyte through a smart device manufactured in the continent at the price of no more than 100 (US dollars) to benefit from all basic e-services and content of which at least 30% is developed and hosted in Africa (*African Union, 2020*).' Since our analyses have revealed that only 40% of people in 33 of the 54 countries in Africa own SPs, these goals may currently be more aspirational than achievable. There are major infrastructural barriers – mainly in rural areas – that will need to be overcome, e.g., extending the electricity grid, increasing cell phone coverage, and expanding bandwidth (*Broadband Commission, 2019*; *Krönke et al., 2022*; *Leo et al., 2015*; *World Health Organization, 2021b*). There are also the inequities in ownership of SPs that we have identified in our analysis (gender, urban/rural residency, age, and wealth) that will also need to be overcome in order to reach the goal of universal access to smart devices. Importantly, there will be a need to ensure sustainability, i.e., that owners of mobile phones can afford to pay for data and continue to utilize their devices. Notably, paying for broadband services in Africa is extremely expensive; in comparison to income, African countries have the highest prices worldwide (*Broadband Commission, 2019*).

Our study has several limitations. First, our analyses are only based on data from 33 out of 54 African countries. We recommend, when/if data become available, conducting the same analyses (as we have conducted here) for the 21 other countries. However, we believe that our qualitative results (i.e., ownership of mobile phones is fairly high, but ownership of SPs is relatively low, and substantial inequities in ownership exist) are likely to be generalizable to those 21 countries. Second, we found that a small percentage of SP owners do not regularly use the internet. While this might limit their availability to participate in many mHealth interventions (unless they were provided with financial means to access the internet), it should be noted that some interventions are available to BP owners, and some can be accessed through shared devices. Third, we have shown that individuals who live in closer proximity to HCs are more likely to own mobile phones. However, we have not demonstrated that owners of mobile phones have access to better healthcare than non-owners. Finally, we have analyzed the most recently available Afrobarometer data (R7), collected in 2017–2018. We plan to analyze data from R8; these data are not yet available. We expect that this future analysis will show that levels of ownership will have increased over the past 3–4 years.

Mobile phone ownership is predicted to keep on growing in Africa, but to what degree this expansion will be in the ownership of BPs or in the ownership of SPs is unknown. Currently, the majority of Africans own a mobile phone, but less than half own an SP. Africa is becoming increasingly urbanized, and urban residents are more likely to be able to afford SPs than rural residents. However, increasing poverty levels throughout Africa – particularly in the African countries where the ownership of SPs is currently low – could limit the expansion in ownership of SPs; ownership of BPs could continue to increase. Some countries (e.g. Botswana) have developed specific plans for digital expansion (*United Nations, 2021*); their plans include strategies for overcoming the current barriers to ownership of smart devices and discussion of the economic, educational, agricultural, and health benefits that could result from such an expansion. Other country-specific plans will need to be developed. As the 'digital transformation' of Africa continues, it will become critical to overcome the current urban-rural, gender, and wealth inequities in mobile phone ownership, particularly, the inequities in the ownership of SPs. If the digital devices needed for mHealth interventions are not equally available within the population (which we have found is the current situation), rolling out mHealth interventions in Africa is likely to propagate already existing inequities in access to healthcare.

## Acknowledgements

We acknowledge Afrobarometer for making available all of the data that were used. JTO, JP, and SB acknowledge the financial support of the National Institute of Allergy and Infectious Diseases, National Institutes of Health grants R56 AI152759 and R01 AI167713. MK acknowledges the financial support of Afrobarometer/the Institute for Democracy, Citizenship and Public Policy in Africa (University of Cape Town). No author or institution at any time received payment or services from a third party for any aspect of the submitted work.

# Additional information

## Funding

| Funder | Grant reference number | Author |
|---|---|---|
| National Institute of Allergy and Infectious Diseases | R56 AI152759 | Justin T Okano<br>Joan Ponce<br>Sally Blower |
| National Institute of Allergy and Infectious Diseases | R01 AI167713 | Justin T Okano<br>Joan Ponce<br>Sally Blower |
| Afrobarometer / the Institute for Democracy, Citizenship and Public Policy in Africa | | Matthias Krönke |

The funders had no role in study design, data collection and interpretation, or the decision to submit the work for publication.

## Author contributions

Justin T Okano, Conceptualization, Data curation, Software, Formal analysis, Validation, Investigation, Visualization, Methodology, Writing – review and editing; Joan Ponce, Investigation, Writing – review and editing; Matthias Krönke, Resources, Data curation, Writing – review and editing; Sally Blower, Conceptualization, Supervision, Funding acquisition, Methodology, Writing – original draft, Project administration, Writing – review and editing

## Author ORCIDs

Justin T Okano http://orcid.org/0000-0002-2476-9370
Matthias Krönke http://orcid.org/0000-0001-8387-9193
Sally Blower http://orcid.org/0000-0003-4342-3911

## Decision letter and Author response

Decision letter https://doi.org/10.7554/eLife.79615.sa1
Author response https://doi.org/10.7554/eLife.79615.sa2

# Additional files

## Supplementary files

• MDAR checklist

## Data availability

All data used in the paper is freely available at: https://afrobarometer.org/data/merged-data.

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
