## [Editor Report]

This study used 2017-2018 Afrobarometer surveys of nearly 45,000 individuals to examine the association between the ownership of mobile phones and proximity to a health clinic in 33 African countries. Findings show that about 40% of people own smartphones and those who live closer to health clinics are more likely to own a mobile phone. This manuscript will be of interest to all people who are involved in the design and implementation of mHealth interventions in Africa.

---

## [Decision Letter]

**Decision letter after peer review:**

Thank you for submitting your article "Lack of ownership of mobile phones could hinder the rollout of mHealth interventions in Africa" for consideration by *eLife*. Your article has been reviewed by 3 peer reviewers, and the evaluation has been overseen by a Reviewing Editor and Bavesh Kana as the Senior Editor. The following individuals involved in the review of your submission have agreed to reveal their identity: Elham Mahmoudi (Reviewer #1); Aurelie Tran (Reviewer #2).

Essential revisions:

1. There are some generalizations that should be considered. For example, it is assumed if participants have access to the internet, they own a smartphone and if they don't, then they are basic phone users. It is possible a lot of smartphone owners do not have subscriptions to the internet due to the high cost of internet in African countries. Can the authors address this?

2. The manuscript mentions inequalities in gender and rural-urban geographic regions based on the odds ratio derived from the BLR. A regression decomposition technique can quantify these differences more elaborately in detail.

3. There is no explanation as to why a lot of poor people own smartphones. This could be due to the use of village phones (first implemented by Grameen Phone in Bangladesh). This has expanded in African countries as well where multiple users communicate through a community phone connecting more users in rural areas. Please address this.

4. Basic phones may also be effective for mobile health interventions through voice-enabled systems and disseminating important messages to communities. (For e.g. there is extensive literature on how community-level messages, such as instructions on personal hygiene and usage of masks, were transmitted through basic phones during the beginning of covid19 in developing parts of Asia). This should be considered.

5. Further clarification of why lack of ownership of mobile phones may propagate inequalities in health is needed beyond just the associations reported.

6. Please present the schematic flow diagram of the final sample size.

7. Describe how missing values were addressed? What proportion of each selected variable (independent or dependent) was missing?

*Reviewer #1 (Recommendations for the authors):*

The analysis is carefully conducted, and the methods section is flawless.

*Reviewer #2 (Recommendations for the authors):*

Perhaps a further discussion of existing initiatives and phone access programs and whether the focus of these programs aligns with the findings in this study could have provided greater contextualization.

*Reviewer #3 (Recommendations for the authors):*

Overall, this article needs substantial revision to be considered for publication.

I recommend the authors conduct a regression decomposition to quantify the differences attributable to gender (men vs woman) or geographical location (rural vs urban).

---

## [Author Response]

Essential revisions:1. There are some generalizations that should be considered. For example, it is assumed if participants have access to the internet, they own a smartphone and if they don't, then they are basic phone users. It is possible a lot of smartphone owners do not have subscriptions to the internet due to the high cost of internet in African countries. Can the authors address this?

We agree with the Reviewer that some smartphone owners may not have access to the internet due to the high cost of internet in African countries. Therefore, to estimate the percentage of SP owners who may not pay to access the internet, we looked at the frequency of access to the internet within this sub-group (Methods: lines 133-138). In the Afrobarometer surveys, participants were asked how often they accessed the internet; they were not asked to specify how they accessed the internet. We analyzed these data, stratified on the basis of the type of mobile phone that we assumed individuals owned (we assumed that an individual owned a smartphone if they reported that their mobile phone could access the internet, and that an individual owned a basic mobile phone if they reported that their mobile phone could not access the internet).

Notably, we found that only 13% of individuals that we classified as SP owners (and 89% of individuals that we classified as owners of BP) reported that they never accessed the internet. We now include the results of this analysis in our revised manuscript (Results: lines 219-221); they are presented in Figure 1—figure supplement 2.

Additionally, we now mention that in order to implement mHealth interventions that are based on smartphones, individuals will need to both own a smartphone and have financial means to access the internet.

2. The manuscript mentions inequalities in gender and rural-urban geographic regions based on the odds ratio derived from the BLR. A regression decomposition technique can quantify these differences more elaborately in detail.

The purpose of our study was to determine – for 33 African countries – what proportion of people owned mobile phones (basic phones and smartphones) in each country, and if there were inequalities/inequities in the ownership of mobile phones based on: (i) gender, (ii) age, (iii) urban-rural residency, (iv) wealth, and (v) distance to a healthcare facility.

We found a high ownership of mobile phone ownership that varies substantially amongst the 33 countries. Additionally, by conducting a Bayesian Logistic Regression we have found that there are significant inequalities/inequities in all five variables. Notably, we have identified substantial differences in the degree of these inequities in the 33 countries.

We agree with the Reviewer that we have not explained why these inequalities exist, and that we could use a regression decomposition analysis to identify explanatory factors. We note that this is the next stage, and current focus, of our research. This next stage requires constructing new statistical models – and utilizing a different dataset – than the models that we present and the dataset that we utilize in our submitted manuscript. Consequently, conducting a regression decomposition analysis is beyond the scope of the present study: it will be an article in its own right.

However, in response to this Comment, we have now added a discussion of potential factors that may explain inequalities in gender and rural-urban geographic regions (Discussion: lines 328-339). These factors have been identified in previous studies.

3. There is no explanation as to why a lot of poor people own smartphones. This could be due to the use of village phones (first implemented by Grameen Phone in Bangladesh). This has expanded in African countries as well where multiple users communicate through a community phone connecting more users in rural areas. Please address this.

We agree with the Reviewer. We now discuss the utilization of village phones in Africa, as well as other explanatory reasons for why a lot of poor people own smartphones (Discussion: lines 339-354).

4. Basic phones may also be effective for mobile health interventions through voice-enabled systems and disseminating important messages to communities. (For e.g. there is extensive literature on how community-level messages, such as instructions on personal hygiene and usage of masks, were transmitted through basic phones during the beginning of covid19 in developing parts of Asia). This should be considered.

We agree with the Reviewer that basic mobile phones may also be effective for mHealth interventions through voice-enabled systems and disseminating important messages to communities. We have added a paragraph (Discussion: lines 370-396) to discuss current mHealth interventions that are being utilized in Africa, including both those based on smartphones and those based on basic mobile phones.

5. Further clarification of why lack of ownership of mobile phones may propagate inequalities in health is needed beyond just the associations reported.

We have added a paragraph (Discussion: lines 356-368) to discuss this topic.

6. Please present the schematic flow diagram of the final sample size.

We have now done so (Figure 1—figure supplement 1).

7. Describe how missing values were addressed? What proportion of each selected variable (independent or dependent) was missing?

We have now described how missing values were addressed, and the proportion of each selected variable that was missing (Methods: lines 118-126 and 147-154).

Reviewer #2 (Recommendations for the authors):Perhaps a further discussion of existing initiatives and phone access programs and whether the focus of these programs aligns with the findings in this study could have provided greater contextualization.

We thank the Reviewer for their comments, and now provide a more extensive discussion of existing mHealth programs as they relate to this study (Discussion: lines 370-396).

Reviewer #3 (Recommendations for the authors):Overall, this article needs substantial revision to be considered for publication.I recommend the authors conduct a regression decomposition to quantify the differences attributable to gender (men vs woman) or geographical location (rural vs urban).

We thank the Reviewer for their comments, and we have now substantially revised the manuscript and addressed all of the points (points 1 to 5) that the Reviewer has raised.